# The Impact of Health and Social Services on the Quality of Life in Families of Adults with Autism Spectrum Disorder (ASD): A Focus Group Study

**DOI:** 10.3390/brainsci12020177

**Published:** 2022-01-28

**Authors:** Cinzia Correale, Marta Borgi, Francesca Cirulli, Fiorenzo Laghi, Barbara Trimarco, Maurizio Ferraro, Aldina Venerosi

**Affiliations:** 1Cooperativa Sociale Agricola Integrata «Giuseppe Garibaldi», 00179 Rome, Italy; cinziacorreale@gmail.com (C.C.); barbara.trimarco@gmail.com (B.T.); autismo@garibaldi.coop (M.F.); 2Center for Behavioral Sciences and Mental Health, Istituto Superiore di Sanità, 00161 Rome, Italy; marta.borgi@iss.it (M.B.); francesca.cirulli@iss.it (F.C.); 3Department of Developmental and Social Psychology, Sapienza University of Rome, 00185 Rome, Italy; fiorenzo.laghi@uniroma1.it

**Keywords:** autism spectrum disorders, parents, health, social, services, quality of life model, social innovation, person-centred care

## Abstract

Background: Autism spectrum disorder (ASD) is a complex developmental condition that affects the whole family. The gap between childrens’ needs and their satisfaction, especially regarding what concerns the presence of social and healthcare services, is still a source of burden, particularly after the transition to adulthood. Our study aimed to gather a comprehensive view on how parents of adults with ASD perceive (and interact with) health and social services, and how the provision of care impacts family quality of life with the aim to advise ASD intervention programs. The goal is to identify specific areas of change useful to influence autism intervention strategies so that they more effectively meet the needs of young people with autism and their families. Methods: We conducted two focus groups with parents of young adults with ASD. A semi-structured focus group methodology was adopted. The QoL conceptual framework guided data collection and analysis as part of a directed theory-driven content analysis approach. Results: The lack of structured care pathways and the low level of integration of different services were the main limits reported by parents during the focus group, while a shared positive perception of the experience conducted together as caring families emerged. Conclusions: The experience here reported claims for a greater role of the institutions in order to facilitate the building of networks that are really inclusive for persons with autism in society and to support the implementation of innovative solutions for the welfare system. Furthermore, parents stressed the need for the provision of support to the family.

## 1. Introduction

Autism spectrum disorder (ASD) is a complex developmental condition that involves persistent challenges in social interaction and communication, as well as restricted or repetitive interests and behaviours. Having a child with autism affects the whole family. It can be stressful, time-consuming, and expensive. Paying attention to the physical and emotional health of the whole family is indeed recommended [1].

Parents of individuals with ASD acknowledge that there is still an important gap to be filled between the need for treatment and its delivery [2], which causes a great burden on the family. International evidence on family’s burden due to ASD highlights a set of aspects which are a source of major concern for parents: the lack of communication with their children, the difficulties of managing behavioural aspects such as aggression and self-harm, the economic burden, the frequent job loss of at least one family member (more frequently the mother), and constraints in social and leisure activities [3,4]. Parents point out that, depending on the age of the child, the gap between needs and their satisfaction increases (that is, greater age and lower satisfaction), especially for what concerns the presence of services aimed to maximise participation in education, employment, and independent living for young adults with ASD [5]. Relations with social and health care professionals become more scattered and ineffective after the transition to adulthood. Contacts with services appear to be more aimed at certifying the need for economic support and less dedicated to tackling the whole and complex clinical-therapeutic and social needs of young adults with ASD, as well as the challenges of daily life [6,7].

In order to put in place a dedicated transition pathway, as well as training tools allowing young people with ASD to be accompanied to adulthood and to reach a fulfilling independent life, a major role should be played by dedicated services with the ultimate goal of favoring a good level of social and working inclusion. However, service delivery systems may vary widely even at the national level (e.g., see [8] for a survey of healthcare services for ASD in Italy) in terms of case management, availability and qualification of health professionals, funding, or level of service integration throughout the transition into adulthood.

Although availability and quality of service can have a major impact on different aspects of life for parents of people with ASD, little is still known about their lived experience (and level of satisfaction) with the services, and how this affects the quality of life (QoL) of the whole family.

Our study aimed to gather a comprehensive view on how parents of adults with ASD perceive (and interact with) health and social services and how the provision of care impacts family quality of life. The goal is to identify specific areas of change useful to influence autism intervention strategies so that they more effectively meet the needs of young people with autism and those of their families. Focus groups were selected as the most appropriate and efficient data collection strategy since group dynamics allow the production of data and insights that would be less accessible without the interaction found in a group. We based our analysis on the domain factors of the QoL model, which takes into account different domains of well-being (e.g., emotional, physical, material, social) and focuses on the person and his/her environment [9,10,11]. Furthermore, we administered a questionnaire aimed at understanding specific aspects of the pros and cons of the parents’ experiences with local services and their suggestions.

## 2. Materials and Methods

### 2.1. Design

In order to explore the experience of families of young adults with ASD requiring substantial support with health and social services, two focus groups were conducted. Focus groups allow participants to develop ideas collectively and construct a shared understanding of the experiences, thus expanding on their own individual priorities and perspectives (Smithson 2008). This is particularly true in focus groups characterized by within-group homogeneity, as in our case, in which participants were all parents of young adults with autism requiring substantial support involved in the activities of an Italian social farm, the cooperative “Giuseppe Garibaldi” (CGG, Rome, Italy). The cooperative represents a case study of a small business enterprise model of social and labour-market inclusion of young adults with autism requiring substantial support directly established by their families. A semi-structured focus group methodology—based on the provision of information/stimulus questions focused on the perceived support from services—was adopted to guide the group discussion. The QoL conceptual framework [11,12] included guided data collection and analysis as part of a directed (theory-driven) content analysis approach [13] (see Section 2.4).

### 2.2. Participants

A purposive sampling, where the cases are not selected randomly but are selected as a group because of the nature of the research question [14], was employed. Parents of young adults with autism (age range: 23–30 yrs) were asked to participate in the focus group discussion. They were recruited via the administrative services of the cooperative CGG, Rome, the lead partner of the project titled “The life-long individualized project: social and labour-market inclusion of young adult with severe autism” (LLIP, founded by the European Social Fund). Inclusion criteria were: willingness to participate in the focus group, being parents of young adults (18–35 years of age) with ASD requiring very substantial support (Level 3, DSM 5), and being involved in the activities of the cooperative. Eight families agreed to participate. Of these, three were represented by mothers, four by fathers and one family participated with both parents. Participants’ characteristics are shown in Appendix A.

### 2.3. Procedure

In order to promote an exchange of opinions and to achieve an in-depth understanding of the phenomena of interest, the eight families were split into two groups (*n* = 4; *n* = 5). Both focuses were conducted following general topics such as experience with health care and social services, support received by health care services, and quality of services for adults with ASD. The semi-structured focus groups took place at the cooperative CGG in Rome, Italy. Encounters lasted about two hours. Following informed consent, parents completed a sociodemographic questionnaire. A researcher with psychology qualifications (CC) facilitated the focus groups. The facilitator was supported by another psychologist (BT) who took field notes. At the beginning of each session, parents were provided with stimulus questions aimed to encourage them to think broadly about their positive and negative experiences with health care and social services. The question fostered initial open discussion between participants, with follow-up questions from the facilitator promoting interaction and discussion. A brief, self-administered questionnaire comprised of 12 questions (11 closed and 1 open) was delivered at the end of the encounter. The questionnaire was aimed at gathering detailed information on the main topic of the focus group (see Appendix A). Of the eight families that participated in focus groups, all filled out the questionnaire.

### 2.4. Data Collection and Analysis

Focus groups were audio-recorded for posthoc analysis of the verbatim transcription. Data analysis was based on a directed content analysis approach [13] combined with the thematic networks analytic technique [15], which consists of using existing theory or prior research as a framework for the coding. The “theory framework” provides “key concepts” or “global themes” that guide the analysis of the text, attributing explicit sentences (proposition) into categories (basic theme). In the current study, the eight-domain factor structure of QoL as developed by Schalock and colleagues [11,12] (Table 1) was used as a conceptual model to guide data analysis as part of a directed (theory-driven) content analysis approach. This model is based on a synthesis of the international QoL literature in education, special education, intellectual disability, mental/behavioural health, and aging and on more recent causal modelling analysis that has evaluated the factor structure and hierarchical nature of the conceptual model [16].

The eight domains of QoL were used to define the coding framework and relationships between codes. Each QoL domain (see Table 1) can be considered a “key concept or global theme” of the qualitative analysis, and each specific QoL’s indicator can be considered a “basic theme” or “category”. In detail, coding the findings involved listing condensed meaning units (proposition) from the two focus groups, identifying the meaningful concepts within each unit, and linking the meaning units to the most appropriate QoL domain. All text that appeared to describe a domain of QoL was highlighted and coded using the predetermined categories (indicators) wherever possible, with a positive or negative signature to indicate the value of the subjective experience. Text that could not be coded into one of these categories was identified and analysed later to determine if they represented a new category or subcategory foreseen by the QoL model. New categories were coded with another label that captured the essence of the QoL domain. The analysis process continued until saturation was reached; that is, there were no new categories extracted from the data. Two authors (CC, AV) independently reviewed all transcripts; disagreements were solved by discussion or by consulting a third reviewer (MB).

As for the analysis of the questionnaire, we collected individual responses and estimated the agreement among participants for each of the issues considered. Consensus level was defined when 62.5% (5 out 8 of parents) of participants answered in the same way regarding the same issue.

## 3. Results

### 3.1. Focus Group

Overall, the coding of the focus group showed that the use of the eight QoL domains as a conceptual model to guide data analysis allowed us to extract a data set able to saturate the parents’ points of view about their experience with health and social services. Only seven of the eight QoL domains were used to code parents’ transcripts. The domain of Physical Well-being, when it emerged during the discussion, was always related to the health of their children and never to their own physical health. Among the domains, the Emotional Well-being, Self-determination, Rights, and Interpersonal Relations domains were particularly quoted. The content analysis allowed us to extract new indicators and sub-indicators not included in the QoL model described in Table 1 that were often shared by parents during the focus group meeting. New indicators that emerged were coded as: Parent Activation and Confidence to Take Action; new sub-indicators were coded as: Right to be Informed, Governance, Integrated Services, Transition to Adult Age, Personal Intervention Program, Case Manager, and Professionals’ Autism Expertise. A number of relationships among the QoL Domains emerged during the text coding (see the section below).

In the following sections, the results will be clustered and described according to the QoL model shown in Table 1. QoL domains and indicators are shown by reporting selected explicit sentences extracted from the verbatim text.

#### 3.1.1. Well-Being

Emotional Well-being

For each participant, we identified and sorted meaning units linking them to the Emotional Well-being domain of the QoL model. In particular, sentences fit with Stable and Predictable, Positive Feedback, and Lack of Stress indicators defined by the model for this domain. The value of sentences was mainly negative, indicating an index of negative experiences in relation with health and social services. Parents personalised their feeling of withdrawal and disinterest from institutions, complaining about the lack of personalised services, remarking on the discontinuity of specific services that should be provided during childhood (i.e., individual assistant at school; transport) and the difficulty of coping with the transition to adulthood. Parents also discussed the lack of a clear and stable point of reference, the lack of planning of available services, and the overall lack of personalised assistance and personalised support planning. Furthermore, parents reported the lack of positive feedback from institutions about their private initiatives to promote autism awareness and social inclusion of their children. Finally, the indicator Safety was represented in the text as parents reported their difficulty to employ competent individual assistants that could substitute them effectively, taking care of their child safely:


*[…] We need to be considered as an endangered species, like pandas. We need to be protected. We need protection. When you start a new path, no one has to question it, it has to be supported, it cannot be uncertain so that I don’t know what happens tomorrow […] Because we families have invested in it, not only in economic terms, but also in terms of my own and my son’s life project.*



*[…] how many neuropsychiatrists have we changed in the last few years? Maybe three? […] and we are dealing with a service that should care for adults with disabilities. It is not possible to have to bear this discontinuity.*



*[Talking about school] […] we have changed two hundred million assistants, for one reason or another they disappeared.*


2.Physical and Material Well-being

Unexpectedly, the Physical Well-being domain was related only to the health of their children. Indeed, parents never talked about their own physical state. Four participants coded the Material Well-being domain in the transcript of the discussion with a negative value. As expected, the identified meaning units were linked to the Financial Status indicator and particularly referred to the out-of-pocket expenditure that families have to undergo to support individual activities and care for their children.


*[Talking about behavioral educational interventions] For these things, we spend a lot of money, a lot of time, a lot of energy.*


#### 3.1.2. Independence

Self-determination and Personal Development

The Self-determination domain was quoted for each family participating in the focus groups except one. Meaning units were attributable mainly to Autonomy/Personal Control, Decisions, Personal Goals and Values, and Choices, all Indicators defined in the QoL model. The value of the meaning units was negative when parents perceived that the institutions raise barriers to their self-determination through lack of services and/or the lack of clear institutional governance specifically aimed at the definition of the individual plan of care. A negative value of the Choice indicator resulted from the hostility or indifference demonstrated by the institution to their suggestions or initiatives. By contrast, the indicators of Choices, as well as Autonomy/Personal Control, Decisions, Personal Goals, and Values, were coded as positive when they reported processes regarding self-directed decisions, implementation of services, choices of specific interventions, or care setting.


*[Talking about the parent’s self-organization in the Cooperative] There are so many who are looking at us with interest, hoping that we succeed because if it turns out that parents are able to self-organize, they can rely on us and help us in this empowerment process.*


Within the Self-determination domain, a novel indicator was created, named Confidence to Take Action, to indicate the positive actions put in place by the families, alone or together, to solve their personal problems, and, as a consequence, generate new relations and/or new services.


*[Speaking about the Cooperative] What if this becomes a model that works. Small cafés can be installed in hotels, in small restaurants, in technical schools so that there can be people [with autism] who can be employed. They have fun with the carpentry workshops. Can you imagine how many situations you could create inside the neighbourhoods? However, this stuff works if it’s a widespread phenomenon.*


As for the Personal Development domain, two parents described that they became ‘experts’ in the field of ASD and the health and social management of their children’s needs. As a result, a new indicator, Parents’ Activation, was coded.

#### 3.1.3. Social Participation

Interpersonal Relations

The Interpersonal Relations domain was coded for every participant with the exception of two. When coded using the Relationships and Support indicators, the value of the meaning units was often negative, indicating the lack of affiliative and inclusive behaviour by social or health operators. Parents reported positive relationships inside the health and social services environment only if there was a personal relationship with the clinician or with the operators (Affiliations, Affection, Intimacy, Friendships).

By contrast, this domain was coded as having a positive value when it was referred to relationships generated through the association with other parents or during the self-promoted initiatives and carried out at the CGG (Affiliations, Affection, Intimacy, Friendships).


*[…] We are demonstrating that when we act as a group, we are a cheerful company, we enjoy working together and we manage to build around us some consensus.*


2.Social Inclusion

As for Social Inclusion, this domain was quoted only when parents talked about services provided to their own children. They never reported any problems relative to their own social inclusion.

#### 3.1.4. Rights

Content analysis of the transcript pointed out the relevance of the Rights domain of the QoL model for all participants. Barriers to accessing services and unmet needs were clearly linked to Human (Respect, Dignity, Equality) and to Legal (Due Process) Indicators as defined by the QoL model for this domain.


*[Related to a sports program] He had to change his clothes between the bushes. Don’t make me say more….*



*[…] Do I have a right to ride the bus or not?*


Analysis of the transcript also demonstrated the presence of new sub-indicators attributed to the Due Process indicator, which is specifically linked to experiences that parents have with services. Among these, there are: the right to have clear and effective information on the services that are mandatory and their local availability (*Right to be Informed*), the right to have stable care across age transitions, and a personalised care plan, and the need/right that institutions play a leadership role in promoting and managing the integration of multiple services by discerning the specific role and economic support that should come from a specific agency (*Governance, Integrated Services, Transition to Adulthood, Personalised Intervention Program; Case Manager*). Finally, among the Due Process meaning units, parents stressed the lack of expertise on ASD in social and health operators, as well as difficulty in the recruitment of personal assistants (Professionals’ Autism Expertise).

#### 3.1.5. Linkages among QoL Domains

Figure 1 describes the main links among QoL domains that emerged during the transcript coding. The links emerged by isolating meaning units that could be associated with more than one QoL domain and coded with more indicators from different domains and/or more indicators of the relative domain, including the new ones. The linkages were found between the QoL domains more represented in the parents’ focus discussion. Emotional Well-being (Stable and Predictable Environments, Positive Feedback), Self-determination (Choice, Generative Capability), and Rights (Due Process) were interrelated in the parents’ experience with services. Some examples showing the interconnectedness among QoL domains are the following:


*[…]*
*We are here suffering, when no one is providing for our children, we try to fill the gap and create a stable care path for them.*



*[…]*
*What is the result? That you as a family are alone in front of the institutions and therefore you succumb.*



*[Talking about the need of support by institution for parents’ self-initiatives]*
*So I hope and dream of a time when the local health authorities present a call for children with autism to self-organize through cooperation.”*


### 3.2. Service Survey Questionnaire

Table 2 reports parents’ perception of their relationship with health and social services obtained through the administration of the Service Survey Questionnaire. The Table shows only those questions for which parents’ agreement above or equal to the threshold of 62.5% (5 out of 8 parents) was obtained. All parents reported a negative opinion of their relationship with health and social services, stressing the fact that the management was poor and disorganised, that there was a lack of clear communication, as well as poor involvement and participation in decisions concerning their children. Administrative issues were the main reason they approached health and social services, claiming a low level of satisfaction while, at the same time, asking for the implementation of projects for independent living as well as job placement.

## 4. Discussion

Very few studies have so far attempted to capture the life-long experience of parents with health and social services using a qualitative approach. Previous studies indicate a low level of satisfaction, strongly emphasising the need for more coordination and support from local authorities, as well as greater involvement of parents in the decision processes [6,7,17]. By contrast, parents often report a good level of solidarity among families and their empowerment in managing daily living problems and directly building enabling and inclusive paths for their children while taking into account their rights [18,19,20]. In the present study, participants described their relationship with health care and social services spanning across the main areas of their children’s life: school, the child and adult psychiatric health units, and the Town Hall services. They described their experiences with these services from several points of view that were easily related to the main QoL Domains.

The concept of quality of life (QoL) has been a useful response to concerns expressed in the field of intellectual disabilities about the need for a framework for service design and evaluation in recent decades. It offers a new way of evaluating disability, providing a useful paradigm that can contribute to the identification, development, and evaluation of supports, services, and policies for individuals with ASD [21]. QoL is being used as the interpretative instrument conceptualized by Schalock, Brown et al. [11] focusing on the person and the individual’s environment. The conceptualization of the quality of life is carried out through the analysis of a number of domains, including well-being, inter- and intrapersonal variability, life environment, a lifespan perspective, holism, values, choices and personal control, perception, self-image, and empowerment. QoL can also be used as a construct to consider how disability affects family behavior and needs, and as a means of assessing quality outcomes and impact evaluation. [22]. At its core, the concept of quality of life gives us a sense of reference and guidance from the individual’s perspective, an overriding principle to enhance an individual’s well-being and collaborate for change at the societal level, and a common language and systematic framework to guide our current and future endeavors. This concept has guided our work with the aim of understanding the family perspective and allowing future policies to be improved, promoting the empowerment and the satisfaction of ASD people and their families. From the QoL analysis and the Service Survey Questionnaire, a coherent picture emerges, testifying to an inadequate offering by the social and health services and the need to be empowered and involved in the co-planning of their children’s individual life projects.

This focus group involved a homogeneous community sharing a common path, starting from school age, with the final aim of finding adequate solutions for their children with ASD. These common objectives have been achieved by implementing original solutions through the establishment of a rural cooperative. Social farming experiences represent an important and expanding phenomenon throughout Europe [23]. Rural areas and urban green spaces can provide opportunities for social engagement and are indeed increasingly recognized for their role in contrasting isolation and loneliness and promoting mental health.

The lack of structured care pathways and the low level of integration of different services were the main limits reported by parents during the focus group. The theme “Transition to Adult Age” was highly quoted in the text, mainly in relation to the weakness of the governance of care pathways, the lack of expertise in planning and managing personalised services by the authorities, the lack of a case manager, and, consequently, the insufficient integration of available supports. The theme of transition to adulthood is especially important in this context, as parents participating in this study represent a small community with this shared interest that emerged when their children’s secondary school ended. Indeed, the transition from youth-based services to adult services can be challenging for individuals with autism and their families. Moreover, social and communication difficulties, narrow interests, and social stigma can negatively affect opportunities for postsecondary education and employment for people with ASD [24]. After having experienced such difficulties and a feeling of “abandonment” from institutions and services, the cooperative Garibaldi was directly established by families as a model of social and job inclusion of their children with autism requiring substantial support. The common goal of these families was to guarantee that their children, after graduating from high school, could move to the adulthood phase having opportunities and resources that could help them build meaningful and connected lives. There is indeed evidence that people with autism, even with a significant need for support, possess capabilities that can be implemented through social inclusion, especially in the school context. In this regard, there are a number of tools that can be used to assist with the development of individualized programs for (young) adults with ASD that can support them in different areas of living, ranging from daily activities to job inclusion [25,26,27]. In this context, it is important to note that there is a lack of research programs on health services for adults with ASD. In particular, there is a need for the identification of comorbid health difficulties, development of new pharmacotherapies, investigation of transition and aging across the lifespan, and consideration of sex differences and of the point of view of people with ASD [1,28,29]. This has a profound impact on the provision of care and planning, especially when it comes to the assessment of each individual’s abilities, needs, and desires [30].

Participants introduced a new aspect inside the domain Self-determination, which is their willingness to co-organise, co-manage, and co-share the planning and implementation of life projects for their children in each life context (school, leisure, home, job….). All of these aspects were coded as the Proactive indicator under the Self-determination domain. Finally, through the coding of the text, the theme of Competence emerged, focusing attention on several levels of ineffectiveness: the lack of expertise of the individual assistant and the inadequate competencies of those involved in ASD care pathways, such as clinicians, public employees, teachers, etc. Specifically, frequent changes in supporting personnel make it difficult for a carer to get to know the person they support and their family, and to understand what works best for them. Furthermore, parents worry about the potential erroneous assessments of their childrens’ needs, which has a profound impact on the provision of care and planning of their future.

The themes that emerged from the discussion reveal how the mostly negative experiences perceived by these parents with health and social services were at odds with a fairly shared positive perception of the experience conducted together as caring families. The positive aspects of this experience concern at least three main domains: Emotional well-being, Interpersonal Relations and Rights. The opportunity of sharing a project as well as the feeling of being neglected by institutions is perceived as a common value. On the other hand, the ‘parental community’ claims the need to have specific rights recognized that have to do with the support of the life path of their children, not so much relating to economic issues, but rather with the lack of recognizable landmarks and of a specific support network to guide the personalised life project sustained on a private ground by the community of families.

The current scientific evidence underlines the need for a lifespan vision to support the transition to adulthood, as well as the requirement to tailor the socio-health services to individual needs. This challenge requires a great deal of innovation in the area of social and health care as well as in civil society itself to avoid the prevailing of bureaucratic reasons over the needs of personal care [31]. Importantly, the relevant points raised as crucial for the management of children with ASD by parents during the focus group are clearly stated by the international recommendations for managing the care of both children, adolescents and adults with ASD. The two most recent European guidelines [1,32], which included a specific analysis of the ASD literature for adulthood, underline the importance of the implementation of integrated services, the presence of a reference person, the provision of support to the family, the provision of specific and individualised support for social inclusion, including employment, and the promotion of awareness in overall society about the unique environmental needs of people with autism. What is therefore claimed by the parents expresses a clear delay in the implementation of international experts’ recommendations.

Furthermore, as described above, there are an increasing number of experiences that see parents as self-directed agents of solution for the management and individual accomplishment of their children in the community in which they live. This experience represents a useful indication for re-thinking the approach of the main social and health care agencies, which are required to drop the prevailing prescriptive behavior and endorse a participatory one. In particular, the use of resources should be revised in order to improve ‘accessibility’ (i.e., giving available and clear information of the paths, available specialized resources in the area of residence), ‘capacity’ (i.e., re-thinking involved figures, training, operationalization of processes), stakeholder involvement (opening to co-production and co-participation) and effectiveness (i.e., improving monitoring of the outcome). Ultimately, the discussion that emerged in the focus groups involved reporting strong disapproval of the current experience of health and social services, requesting the providing agencies to take on a leadership role, thus recognizing the intrinsic value of public institutions and care bodies while highlighting that currently, they do not achieve the level of quality required by the community they serve. Clearly, there are very specific geographical differences that need to be taken into account. In countries such as the UK, not-for profit advocacy organizations play a very important role in collecting family voices and needs, as well as coordinating services and opportunities for life planning. In Italy, by contrast, charities and advocacy groups are not well-organized as a network and cannot join forces [20].

The present study includes some important limitations due mainly to the small number of families involved and the lack of a comparison group. Future research should compare the point of view of parents that have experienced different kinds of service solutions, i.e., self-directed vs. community or residential services. Moreover, future research should consider performing QoL-grounded focus groups with the stakeholders involved in the support of youngs and adults with ASD in order to gain a multi-perspective analysis of the context and, hopefully, to simulate a mutual understanding and a shared cluster of solutions.

## 5. Conclusions

The experience here reported calls for institutions to engage in greater support and promotion of these initiatives in order to facilitate the building and support of networks that are functional to fully include persons with autism in society. In the era of the rise of participatory methods applied in several societal sectors, including social coexistence and inclusion [33], and socio-economic sustainability of the communities of the territories [34], the experience reported by this group of parents appears emblematic beyond the specificity of ASD [35].

## Figures and Tables

**Figure 1 brainsci-12-00177-f001:**
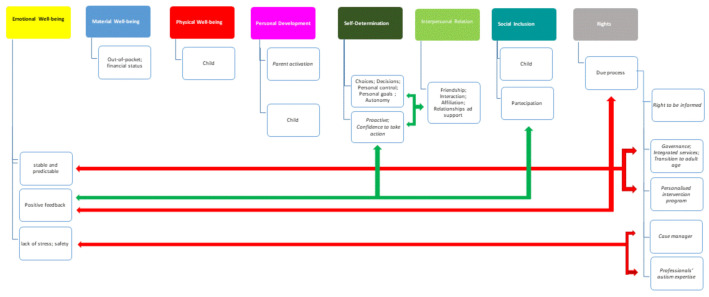
Linkage among QoL domains more represented in the parents’ focus discussion on their experience with health and social services. Colored boxes represent the eight domains of the QoL theory. White boxes represent the QoL indicators confirmed by the coding of the meaning units, and the new indicators or sub-themes (italics) that emerged by content analysis. Rows are positive (green) and negative (red) relationships between themes, as extrapolated by content analysis. Their thickness indicates the frequency of quotation in the text.

**Table 1 brainsci-12-00177-t001:** Eight-domain factor structure of QoL (modified from [11,12]).

Factor	Domain ^a^	Indicators ^b^
Well-being	Emotional well-being	Contentment, self-concept, lack of stress, safety, stable and predictable environments, positive feedback
	Physical well-being	Health and health care, mobility, wellness, nutrition, activities of daily living, leisure
	Material well-being	Financial status/possessions, employment, ownership, housing
Independence	Personal development	Education and habilitation, personal competence, performance, purposive activities, assistive technology
	Self-determination	Autonomy/personal control, decisions, personal goals and values, choices
Social participation	Interpersonal relations	Interactions, relationships and supports; affiliations, affection, intimacy, friendships
	Social inclusion	Community integration and participation, community roles, social/natural supports, integrated environments
	Rights	Human (respect, dignity, equality) and legal (citizenship, access/barrier-free environments, due process, privacy, ownership)

^a^ QoL core domains represent the range over which the QoL concept extends and thus define the multidimensionality of a life of quality; ^b^ QoL indicators are QoL-related perceptions, behaviours, and conditions that operationally define each QOL domain [12].

**Table 2 brainsci-12-00177-t002:** Family’s perception of the relationship with health and social services. Only questions that showed parents’ agreement above or equal to the threshold (62.5%, 5 out of 8 parents) are shown.

Questions	% of Parents Agreeing with a Specific Option
What are the positive aspects of your Local Healthcare Authority?	100% Few or none
What are the negative aspects of your Local Healthcare Authority?	62.5% Extended waiting times; 75% Unclear communication; 87.5% Poor service offer; 75.0% Disorganization of the service; 62.5% Difficulty in accessing services; 75.0% Poor involvement and participation in decisions related to your child
What are the reasons for approaching your Local Healthcare Authority services?	87.5% Certifications/bureaucratic issues
Which of your Local Healthcare Authority services would you like to see implemented/improved?	62.5% Projects for independent living/cohousing/protected apartments; 75% Family support; 87.5% Job placement paths
Overall satisfaction for your experience with healthcare authority in the last 12 months (Likert 1–10 from ‘bad experience” to “great experience”)	73% Low satisfaction
What are the reasons for approaching your Town Hall services?	62.5% Recreational and sports projects and workshops
Overall satisfaction of your Town Hall authority (Likert 1–10 from “bad experience” to “great experience”)	75% Low satisfaction

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
