# Peer review of "The Impact of Health and Social Services on the Quality of Life in Families of Adults with Autism Spectrum Disorder (ASD): A Focus Group Study"

_brainsci, 2022, doi:10.3390/brainsci12020177_

Round 1

Reviewer 1 Report

The authors present findings of focus groups with parents of adult children with ASD.  The sample is a homogeneous group of individuals with the authors' acknowledge.  The QoL framework is important and beneficial when applied to understanding the experiences of autistic adults and their families.  At times, however, it is unclear whether the findings are intended to support research and changes to resources for autistic adults or whether the purpose is to support parents and family members of these individuals.  Clarity regarding intentions and outcomes would strengthen the manuscript.  The manuscript would also be strengthened by a discussion of the limitations of the study design and homogenous sample of participants.  What are the next steps for the research agenda?

Additional formatting is needed as there is a change in how citations are done as well as inconsistent capitalization of key terms related to QoL.

Author Response

Response to Reviewer 1 Comments

The authors present findings of focus groups with parents of adult children with ASD.  The sample is a homogeneous group of individuals with the authors' acknowledge.  The QoL framework is important and beneficial when applied to understanding the experiences of autistic adults and their families.  At times, however, it is unclear whether the findings are intended to support research and changes to resources for autistic adults or whether the purpose is to support parents and family members of these individuals.  Clarity regarding intentions and outcomes would strengthen the manuscript. 

We appreciate the reviewer’ suggestion and included a sentence both in the Abstract and Introduction to clarify the purpose of the study and what findings could implicate to support research and changing to resources for autistic adults and their families. The revised text is underlined in yellow.

The manuscript would also be strengthened by a discussion of the limitations of the study design and homogenous sample of participants.  What are the next steps for the research agenda?

 Thanks to the reviewer for pointing that out and giving us the chance to say the limitation of study design and our indication for the next steps for the research agenda. Please see in the discussion we included a new paragraph (rows 447-454) to address this points.

Additional formatting is needed as there is a change in how citations are done as well as inconsistent capitalization of key terms related to QoL.

We proofed checked and revised to harmonise capitalisation of key terms related to QoL.

Finally, some rephrasing was done to improve English. The revised text is underlined in yellow.

Reviewer 2 Report

This is a very well-written and presented paper on an important research topic. I commend the authors. I thought the use of the QoL model was well linked to the study design and presentation of the findings and the discussion and conclusions were appropriate. Figure 1 is a particularly helpful image for conveying the findings. 

Author Response

We thank you very much for the positive comment received from our study and for appreciating the use of the Qol model as a framework for interpreting the collected data.